# Worldwide Research on Circular Economy and Environment: A Bibliometric Analysis

**DOI:** 10.3390/ijerph15122699

**Published:** 2018-11-29

**Authors:** José Luis Ruiz-Real, Juan Uribe-Toril, Jaime De Pablo Valenciano, Juan Carlos Gázquez-Abad

**Affiliations:** Faculty of Economics and Business, University of Almeria, Ctra. De Sacramento, s/n, 04120 Almería, Spain; juribe@ual.es (J.U.-T.); jdepablo@ual.es (J.D.P.V.); jcgazque@ual.es (J.C.G.-A.)

**Keywords:** circular economy, environment, bibliometric analysis, WOS, sustainability, cross world research, global research environment

## Abstract

The relevance of circular economy to environmental science has led to a notable increase of research works during the last few years. It is very important to know the evolution of the publications that relate these two concepts, as well as the main areas of knowledge in which these investigations are framed. The purpose is to understand and highlight the state of art of circular economy and the role and relationship of the environment. Bibliometric analysis allows to evaluate developments in knowledge on a specific subject and assesses the scientific influence of researches and sources. This paper analyses the worldwide research dynamics on circular economy in the period from 2006 to 2017. A bibliometric analysis of 743 articles was completed. The most productive journals in this field were Journal of Cleaner Production. The five most productive countries were China, United Kingdom, Italy, the Netherlands, and Germany. Works on the circular economy and environment has considerable potential and it is open to research fields as sustainability or industrial production. The findings of this study could prove useful for studies into environmental circular economy, as they show a global sight of this line of study. Thus, the article represents a contribution to identify the main trends in circular economy research and environment and, from there, propose future research initiatives.

## 1. Introduction

Circular Economy (CE) is a topic that has been gaining importance and prominence in recent years, both among scholars and managers of public and private entities, having become trending in a few years. A proof of this is the rapid growth in the number of publications on the topic in just a few years. Thus, for example, the number of academic articles published on CE and environment was only 27 in 2014, reaching 371 in 2017, which represents an increase of 1275% in just three years.

In addition to being an upward topic in western economy, CE is having a rapid growth in other countries, mainly in China, which is the first country in the world to develop a law (in 2008) that facilitates the implementation of CE [1]. In a socioeconomic context of business development, CE is considered an important approach to achieve sustainable economic and environmental development [2]. Thus, the European Union also created a CE package by extending the earlier waste directive [3].

However, given its transversal and multidisciplinary nature, there is no single definition of CE, having been defined in numerous ways and having a different meaning for different audiences and contexts. In this sense, the authors of [4] even argue that there is no commonly accepted definition of CE. In any case, it is clear that the concept of CE has evolved in recent years. Circular Economy is a mode of economic development whose purpose is to protect the environment and prevent pollution, thus facilitating sustainable economic development [5]. Singh and Ordonez [6] understand CE as an economic strategy to promote innovative ways to transform the current linear consumption system into a circular one, through material savings. CE is a generic term covering all activities that reduce, reuse, and recycle materials in the process of production, distribution, and consumption [7,8]. Other authors [9] conceive CE as a production and consumption system with minimal losses of materials and energy through extensive reuse, recycling, and recovery. The results of the analysis of 114 definitions [10] indicate that CE is most frequently depicted as a combination of reduce, reuse, and recycle activities. In the same research, these authors define CE as “an economic system that is based on business models which replace the ‘end-of-life’ concept with reducing, alternatively reusing, recycling and recovering materials in production/distribution and consumption processes, thus operating at the micro, level and macro level, with the aim to accomplish sustainable development”.

Some authors identify the fundamental defining dimensions of a CE and frame them for interdisciplinary research such as built environment [11], building industry [12], or family businesses [13], describing the development of the CE scale treated as an instrument that allows for a direct measurement of the importance of CE for companies.

There is also a rapid growth in the number of consultancy reports, driven by greater demand from clients [14], which is mainly due to the fact that CE is perceived by companies as a way to be able to implement sustainable development [15]. Although sustainability is based on three dimensions from a holistic perspective: environmental quality, economic prosperity, and social equity [16], most of the literature maintains that CE mainly focuses on environmental quality (e.g., [17,18]). In this same sense, the authors of [19] maintain that CE as an approach to combat environmental challenges and promote sustainable development has recently received increasing attention. The Circular Economy and Environment (CEE) unit help to concretize and direct the transition to a sustainable society.

As precursors to the environmental economic analysis, the authors of [20] argue that the concept of CE, addresses the interlinkages of the four basic welfare economic functions of the environment: (1) amenity values; (2) a resource base for the economy; (3) a sink for residual flows; (4) a life support system. Environmental economic analysis offers an analytical approach and provides a whole description of the environmental consequences of various choices in public policy-making, thereby making the analysis interdisciplinary [21,22]. Neoclassical environmental economic analysis emphasises on the utility of the environment, as measured in terms of economic welfare, thus becoming a fundamental life support system [21].

For all these reasons, we consider that it is very important to know the evolution of the publications that relate these two concepts, CE and environment, as well as the main areas of knowledge in which these investigations are framed. The purpose of this research is to understand, from a bibliometric perspective on scientific literature, the state of art of CE and the role and relationship of the environment. Bibliometric analysis allows to evaluate developments in knowledge on a specific subject and assesses the scientific influence of researches and sources [23]. Thus, we carry out a bibliometric analysis, as detailed in the next section. Next, the results obtained in this research are presented in relation to the countries and languages of publication, the annual evolution of the number of publications, the most influential journals and authors, as well as the main areas of knowledge in which the publications are framed.

## 2. Methodology

In order to study the CE evolution in scientific publications and its relationship with the environmental context, given their growing influence in academic literature, a bibliometric analysis was carried on. This bibliometric study is based on a systematic bibliographical analysis of the literature related to the central study theme, following a sequence of steps [24]: (1) define the search criteria, keywords, and time periods; (2) selection of Web of Science database; (3) adjustment and refinement of research criteria; (4) full export of result; (5) analysis of the information and discussion of the results (Figure 1).

Thus, circular economy and environment were the terms selected and the research scope focused on articles published in the period 2004–2017 (Thus, excluding publications of 2018, year for which there has not yet been enough time for the works to have been cited at the moment in which the research has been developed). The following was identifying publications from a robust and reliable database. Garfield [25] first described a citation index for science. The publication of indexes in the Web of Science (WOS) online database core collection was considered [26]. Despite the limitations of using a single database, WOS provided enough information for the purpose of this research. WOS is a multidisciplinary database which mainly records scientific articles, reviews, and books, but also other documents such as meetings, editorials, or letters. Although there is a close correlation between several bibliometric indicators and the database [27], WOS was considered the best option due to its quality, the possibility to search and filter search using several bibliographic parameters, provides easy access to the full texts of the searched papers, and that it is the most commonly used database, generating useful information for researchers evaluating scientific activity.

The preliminary results of this search in WOS, without applying any filters, were a total of 1435 documents. These results were adjusted and refined, according to the search criteria defined for this research. So, only Web of Science Core Collection was selected, obtaining 1409 documents. This WOS Core Collection was subsequently filtered, redefining the date until 2017 (993 documents), and filtered to only include articles published in scientific journals. This facilitated the research, since besides guaranteeing the quality of the publications, the journals include multidimensional elements, such as citation, time, language, etc. [27]. After debugging the database, the initial query on these terms in the titles, abstracts and keywords resulted in 743 documents.

Once we had the final data, we exported the results with all available information in “.txt” format, which we used later for the bibliometric analysis. For the publications identified according to the criteria described above, the following elements were considered: number of annual publications, language, countries, journals, authors, and areas of knowledge. Bibliometric analyses are mainly based on two criteria: the scientific publication, as an indicator of research output [28], and citations received by them as a proxy of their scientific impact [29]. Thus, different bibliometric indicators were also used in this research to characterize the scientific output: the impact of papers, indicated by the number of references received from other subsequent publications (number of citations); frequency: Hirsch index (h-index and averages), proposed by [30] and defined as the number of papers with citation number ≥h; and the impact factor of journals in the Journal Citation Report^©^.(JCR—Quartiles).

## 3. Discussion of Results

Circular Economy is a very specific and emergent term and, although it is a concept that is related to multiple topics (e.g., sustainability, engineering, ecology, cleaner production, eco-efficiency, closed loop economy, zero waste economy, etc.), for this research we wanted to focus on the scientific publications that connect it with the environmental element. Therefore, the keywords search focused on the expression “circular economy”, as well as the word “environment” and its derivative “environmental”.

### 3.1. Most Influential Countries and Languages

China leads the ranking of the most influential countries (Table 1), both in number of publications (21%) and citations (23.8%), followed by the United Kingdom with 102 articles (10.6%). This may be due to the great importance that these topics have in China in recent years, not only in the academic and research fields, but also because of the sustainable development strategy proposed by the central government of China, aiming to improve the efficiency of materials and energy use [31]. Figure 2 shows the number of publications of each country. However, considering other indices, such as total citations per article and the h-index, the ranking undergoes significant changes, with countries such as Norway, the United States, South Korea, Japan, Canada, or the Netherlands, acquiring a leading role. The striking case of Norway, with an average of citations per article of 65.54, is due to the high number of citations of just two papers about China’s growing CO_2_ emissions [32] and product services for a resource-efficient and circular economy [33] (354 and 222 respectively). In any case, the first quartile of countries in the ranking encompass two-thirds (66.07%) of the total citations (Figure 2).

In addition, Figure 3 shows, in a visual way, the geographical distribution of citations of articles, with a density map.

Although the country that dominates the publications on the topics analyzed in this research is China, the most frequent language of publication is English (Table 2), representing 98.6% and other languages being almost irrelevant. This is because the most relevant journals on these issues publish in that language.

### 3.2. Evolution of the Number of Publications per Year

The historical development of published articles containing the terms circular economy and environmental in the title, abstract, or keywords in the Web of Science is shown in Figure 4. The first article on this subject is relatively recent, published in 2006: “Education for regional sustainable development: experiences from the education framework of HHCEPZ project”, published in Journal of Cleaner Production.

During the first years, the growth of publications addressing this topic was moderate, reaching 35 articles in 2012. From there, and after two years of certain stagnation in the number of annual publications (with only 27 papers in 2014), in 2015 again shows a growth, but this time much more intense, with 63 papers in 2015, 172 in 2016, and 371 in 2017, which represents a growth of 112% with respect to the previous year. This evolution, mainly in recent years, shows the importance of this topic in the literature, having become an emerging trend.

Regarding the evolution of the number of citations (Figure 5), its evolution is similar to the number of annual publications, with some slight differences. In any case, it seems evident that the evolution also follows an ascending line, although the articles published in 2017 have barely had time yet to have a number of citations as high as those published in previous years.

### 3.3. Most Influential Journals

Publications related to circular economy and environment can be found in a wide range of journals and different areas of knowledge. There are more than one hundred journals that have published articles on this subject, which indicates their importance in the academy, especially in recent years. By focusing the analysis on journals whose publications represent at least 1% of the total on these topics, then 17 journals can be ranked (Table 3).

The leader of this ranking is Journal of Cleaner Production that besides having the largest number of publications (140) on these issues, representing 18.8% of the total, has also the highest h-index (28), what seems to indicate that this journal has studied these topics in depth.

Only five other journals have published articles that represent more than 4% of the total with respect to the topics analyzed in this research: Sustainability (5.2%), Advanced Materials Research (4.8%), Resources Conservation and Recycling (4.8%), Procedia CIRP (4.4%), and Journal of Industrial Ecology (4.1%).

Attending to h-index and number of citations, apart from Journal of Cleaner Production, already mentioned and clearly leading the number of citations (3439), there are some more journals that it is worth highlighting, such as: Resources Conservation and Recycling (13 h-index; 645 citations), Journal of Industrial Ecology (12 h-index; 607), and Sustainability (9 h-index; 253). A different case is that of the journal Renewable Sustainable Energy Reviews, as it has only 8 articles, but an h-index of 7 and a relevant number of citations (277). This is due to the high number of citations of the article “Recycling of WEEEs: An economic assessment of present and future e-waste streams” [34], which has been cited 132 times in the WOS Core Collection.

### 3.4. Most Relevant Authors and Cited References

The bibliometric indicators are based on the idea that the impact of a paper can be measured by counting the number of other papers which make reference to or cite it. This study differentiates between papers and author citations.

Thus, upon analyzing the 743 articles published during the period 2006–2017, this research identifies 3300 different authors (After debugging repetitions of authors’ names), with a range of publications between 1 and 38 articles. The top ten most active scholars (Table 4) have produced 14.67% of the papers. According to the data, this research reveals that Professor Yong Geng, from Shanghai Jiao Tong University, is the most influential author, since their publications about the topics analyzed have been cited 1484 times, with 19 h-index. Professor Geng is also the most prolific author on the subject, with 38 articles. Thus, his average citation per paper is 39.05. There are other relevant authors in this topic, such as Tsuyoshi Fujita (11 h-index, 12 articles), Sarkis (10 h-index, 10 articles), Zhu (9 h-index, 12 articles), and Liu (8 h-index; 10 articles), among others. A remarkable case is that of Professor Ulgiati, who with 7 articles published has 445 citations, which represents an average of 65.57 per paper. This is mainly due to the 225 citations of the paper “A review on circular economy: the expected transition to a balanced interplay of environmental and economic systems”, published in 2016, in Journal of Cleaner Production. Figure 6 shows a cocitation map based on bibliographic data. The minimum number of documents of an author has been stablished on three articles. Of the 2154 authors, 118 meet the thresholds. For each of the 118 authors, the total strength of the co-authorship links with other authors has been calculated. Some of them are not connected to the other ones, so they are not shown on the graphic. The largest set of connected authors consist of 64 links. Eleven clusters were found, highlighting cluster in yellow, integrated by six authors: Geng, Lai, Sarkis, Ulgati, Zhang, and Zhu.

The most influential authors from each cluster can be easily identified in most of the groups. The leader of cluster 1 (red) composed by twelve authors is professor Wang, and in cluster 2 (dark green), with eight authors, is Fujita. Both are the most crowded groups. Six authors are shown on cluster 3 (dark blue), cluster 4 (yellow), and cluster 5 (purple) lead by Wang, Geng, and Xue, respectively. 

Liu is in front of group 6 (light blue), professor Li in cluster 7 (orange) and knot 8, in brown, Chen. The smallest groups surrounding the centre of the graphic are cluster 9 (pink), cluster 10 (light pink), and cluster 11 (light green), have no leader.

### 3.5. Main Areas of Knowledge

Given the transversality of the topics under study, after the initial analysis this research sought to know in what areas of knowledge the articles related to circular economy and environment have been published, and in which of them they are becoming an emerging source for the academy. 

There are a wide variety of areas of knowledge dealing with this subject (Figure 7), and obviously, many of the articles analyzed correspond to more than one area of knowledge. The most prominent field is Environmental Sciences Ecology (highlighted in red), with 645 articles. That is, 86.8% of the papers published on these topics fall within this area of knowledge. The second in importance is Business Economics, with 592 articles, which represents 79.6% of the total, and the third field is Engineering, with 552 papers (74.3%). Therefore, each of these three areas of knowledge includes more than three-quarters of the total articles analyzed, being clearly the most relevant fields. Other areas of knowledge complete the top ten ranking: Energy Fuels (44.9%), Science Technology (41.4%), Computer Science (40.6%), Public Environmental Occupational Health (17.9%), Materials Science (16.9%), Social Issues (16.7%), and Agriculture (12%). 

Figure 8 shows a map based on bibliographic data on co-occurrence on the authors keywords from 2014 to 2017 using a fractional counting method. The minimum of occurrences of a keyword was established on five of the 1254 keywords. This map is useful to know trends on researching about CE.

Nowadays, eco-innovation, eco-design, and waste management [35] are the topics were academia is focusing. Subjects related with industrial (industrial ecology, industrial parks or cleaner production) have been left behind but still have a high average on citation. 

The map also shows the main interactions between the most frequent terms in this research, and the existing clusters, highlighting the term circular economy as the central figure. There are seven clusters: the first one related to energy and industry (including China as keyword). In the second, there are relevant keywords such as remanufacturing or resource efficiency. The third one is about eco-efficiency and cleaner production. The fourth cluster is related with life cycle assessment, also known as LCA. Three items compose the fifth group: eco-innovation, green economy, and sustainable development. The last two clusters, climate change and environment, are at the edge of the map, with only one item each.

## 4. Conclusions

The concept of CE has evolved, becoming for a few years a trending topic, with increasing relevance, both at the level of management of public administrations and companies, as well as at the academic level. Governments, like consumers, are increasingly aware of environmental care and see in the CE both a philosophy and a strategy that can promote clean growth and improve environmental conditions. Thus, for example, governments are imposing new restrictions on pollution and waste that apply along entire product life cycles [36]. In this sense, it is noteworthy, among other measures taken by public administrations of different geographical areas, the proactivity that China and the European Union are having in the development of legislation that develops and consolidates the implementation of the CE in their territories. Despite this, it is still necessary to define better the areas and sectors that fall within the scope of CE, because there is still way to go. Currently CE is impregnating the management of companies in very diverse activities and productive sectors, such as recycling and reuse in the plastics economy, innovative models for waste management in emerging markets, CE in food, fashion, design, or the development of innovative products for a CE.

Along with this development of the CE in the public sector and business management, there is also a growing among the academies, with numerous publications in very diverse areas of knowledge, given the multidisciplinary and transversal nature of this field, and also because the interpretation and application of CE have been very diverse. Although CE has in its origin an orientation towards three fundamental pillars, the impact on economy, society, and the environment, is in this last area where CE has focused the most, both in management and scholar publications. Among other reasons, this may be because this field covers different and very diverse subsections and specialties and its results are quite visible. Even in the early work on the concept of CE, close ties were already established with the environment, as in “Economics of Natural Resources and the Environment”, published by Pearce and Turner in 1990 [20]. So, Circular Economy (CE) and Environment (E) are closely linked in the academy, even configuring themselves as a new space for the development of future research, as CEE (Circular Economy and Environment), as a formula for manage the transition to a sustainable development.

All the above are some of the reasons why we consider of great interest to know the evolution of the publications that relate these two concepts (CEE), as well as the main areas of knowledge in which these investigations are framed. Thus, the main purpose of this research is to understand, from a bibliometric perspective on scientific literature, the state of art of CE and the role and relationship of the environment. For this, we carried out a bibliometric analysis, focus on countries and languages of publication, the annual evolution of the number and citations of publications, the most influential journals and relevant authors, as well as the main areas of knowledge in which the articles are framed. To this end, in addition to quantitative indicators, we have included other indicators of a qualitative nature, which provide us with a much broader perspective on the subject matter of the research. This work may also help researchers to develop more specific CE bibliometric analysis to areas other than environment, just analyzed in this research.

This research allows us to observe how, despite the fact that China is the country with the highest number of publications on CEE (21%) and citations (23.8%), with an important difference with respect to the second one (United Kingdom), English is the reference language of publications in this field (98.6%). It is noteworthy that certain countries, that are large economies and greenhouse gas emissions, with evident potential interest in CE management and its relationship with the environment, hardly occupy a relevant place in the number of publications on the subject. This is, for example, the case of Russia, not generating hardly publications in this research field. Another remarkable case is that of South America where, apart from Brazil, no other country appears in the top 25 countries in the ranking. Even Brazil hardly contributes with 16 articles on this topic, very far from what could be expected considering its economic and geographical dimension, as well as its interest in the subject.

Considering the historical evolution of publications on this topic, it can be seen that the first article is quite recent (2006). The pace of publications was moderate during the first years, until 2015, the year after which this subject acquires a relevant role, and the publications soar until reaching 371 articles on CE and environment in 2017. This is clear evidence of the current importance of this topic. Attending the number of citations, it presents a very similar time evolution, with the exception of the year 2017, given that these publications are very recent, with hardly any time yet to have a high number of citations. This also shows that the topic is emerging among scholars. The number of articles published during the period 2006–2017, was 743.

In order to offer a more complete analysis, this bibliometric research analyzes both quantitative and qualitative indicators. Thus, in the case of the most relevant authors, this study differentiates between papers and author citations. This research identifies 3300 authors, Yong Geng (at Shanghai Jiao Tong University) being the most prolific (38 articles) and the one with the most citations (1484 times, and 19 h-index). There are some striking cases, such as the paper “A review on circular economy: the expected transition to a balanced interplay of environmental and economic systems” [15], published by Ulgiati (2016) in Journal of Cleaner Production, with 225 citations.

Attending the main journals and areas of knowledge in which this topic has more relevance, publications can be found in a wide range and different fields. More than one hundred journals have published articles on this subject, of which we highlight those that represent at least 1% of the total of publications, finding 17 journals. The journal with the highest number of publications on this topic (140 articles, which represents 18.8% of the total), citations (3439), and highest h-index (28), is Journal of Cleaner Production. Others relevant journals by the number of publications in this subject are: Sustainability, Advanced Materials Research, Resources Conservation and Recycling, Procedia CIRP, and Journal of Industrial Ecology. Focusing on h-index and the number of citations, the more relevant journals are almost the same: Resources Conservation and Recycling (13 h-index; 645 citations), Journal of Industrial Ecology (12 h-index; 607), and Sustainability (9 h-index; 253).

Although there are many areas of knowledge that are interested and involved in such a transversal concept as CE and its relationship with the environment, and there are a wide variety of them dealing with this subject, this research identifies three well-highlighted areas of knowledge: Environmental Sciences Ecology (645 articles, that is 86.8% of the papers published on these topics), Business Economics (592 articles, 79.6%), and Engineering (552 articles, 74.3%). Thus, it can be said that this topic is yet an emerging source for the scholars in general, and for these research fields specifically. 

Through this bibliometric analysis we have detected that numerous researches have focused on specific processes within the CE, such as the 3 Rs of the environment: reduce, reuse and recycle, and some others, with a broader approach, on more general aspects or conceptual models, such as the main principles of CE. We consider of great interest to investigate and deepen the analysis of specific cases (case studies) of success in public programs of some governments, as well as in relevant business strategies, as a way to provide benchmarking. On the other hand, many of the works focus only on the relationship of the CE with the environment. Although this is very important, it leaves aside fundamental aspects of the CE, such as the social and economic impact. The development of research with a holistic approach would be desirable.

By using fractional counting method, different trends have been identified: eco-innovation, eco-design, and waste management. Other interesting interactions between the most frequent terms are those related to energy and industry, remanufacturing and resource efficiency, eco-efficiency, and cleaner production. Topics such as LCA, green economy, sustainable development, climate change, and environment are also emergent research lines.

This work is not exempt from certain limitations, some of which could be the basis for future research. Some of them are derived directly from the characteristics and shortcomings of the bibliometric analysis, which is a method of quantitative analysis by nature. However, quantitative metrics are weak choices for assessing the research output of an individual scholar. Some authors may publish only a few articles, but they are very influential in their area of knowledge, even being seminal for a specific field, or having a great impact on it. This is why we also measure qualitative features and standardized metrics, such as the number of citations, or the h-index. The full identification of the impacts produced by scientific publications should include both directly identifiable research impacts (for example, citation counting), and indirect impacts among scientists, what can be achieved with dating mining developed by [37]. Anyway, the methodology of this research could be expanded with other quantitative or qualitative tools (e.g., knowledge maps or visuals).

Although the WOS database has been used in this paper, since it is the most influential tool for a bibliometric analysis, other instruments could be considered in the future, such as Scopus, or Google Scholar.

Finally, this research has focused on publications where the concepts of CE and environment are related, but with a generic character. For future researches on the bibliometric analysis of these terms, the focus could be addressed to different fields of this discipline, including and analyzing more specialized documents. It could be also interesting to implement a systematic literature review on CE using others methods and tools, such as meta-analysis, that may provide a different point of view.

## Figures and Tables

**Figure 1 ijerph-15-02699-f001:**
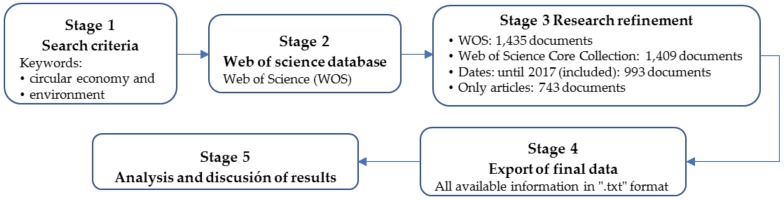
Five stages of bibliometric analysis.

**Figure 2 ijerph-15-02699-f002:**
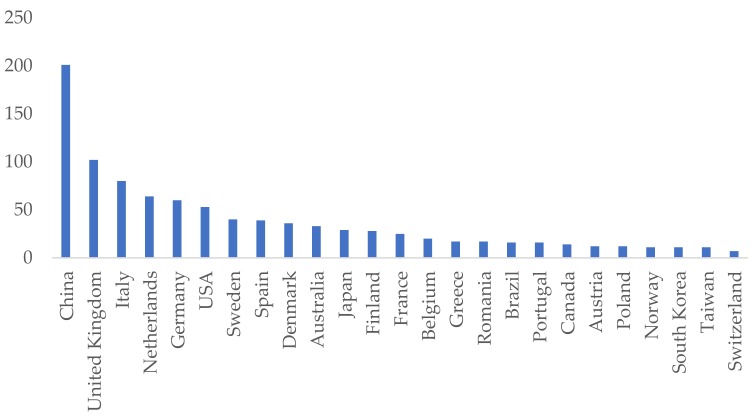
Articles per country.

**Figure 3 ijerph-15-02699-f003:**
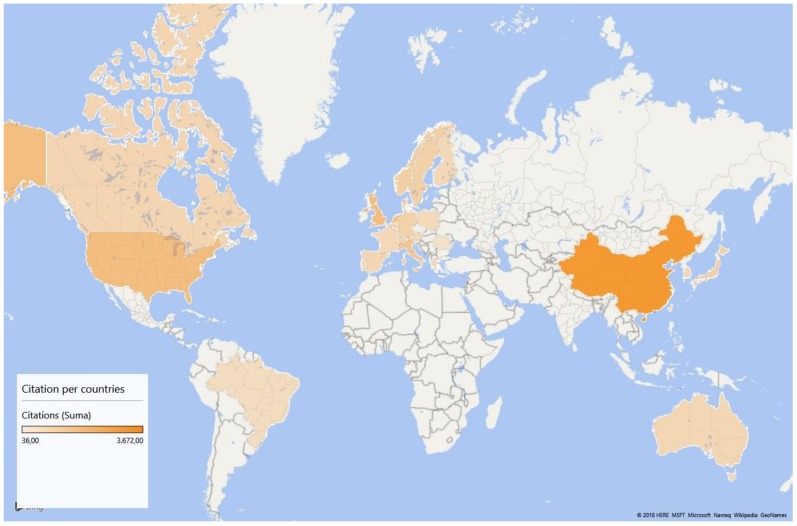
Density map according to citations.

**Figure 4 ijerph-15-02699-f004:**
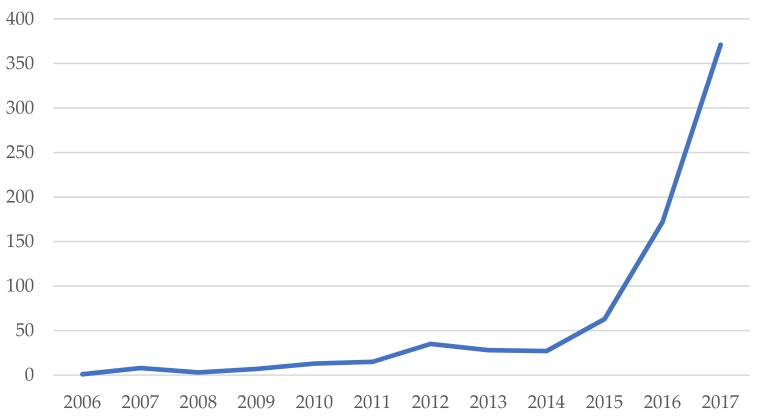
Annual evolution of publications.

**Figure 5 ijerph-15-02699-f005:**
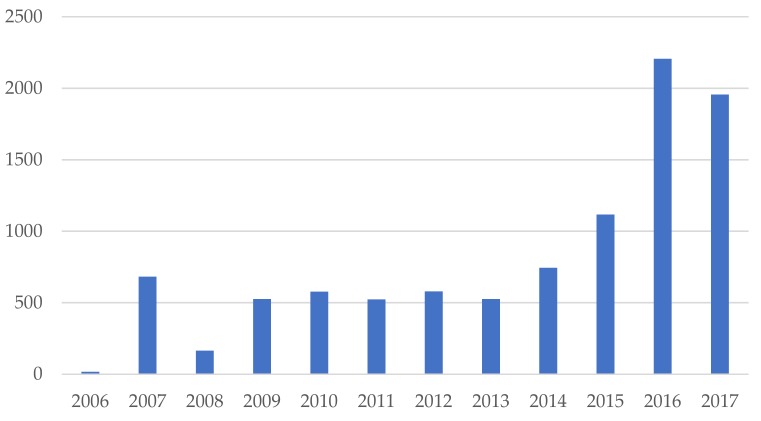
Number of citations per year.

**Figure 6 ijerph-15-02699-f006:**
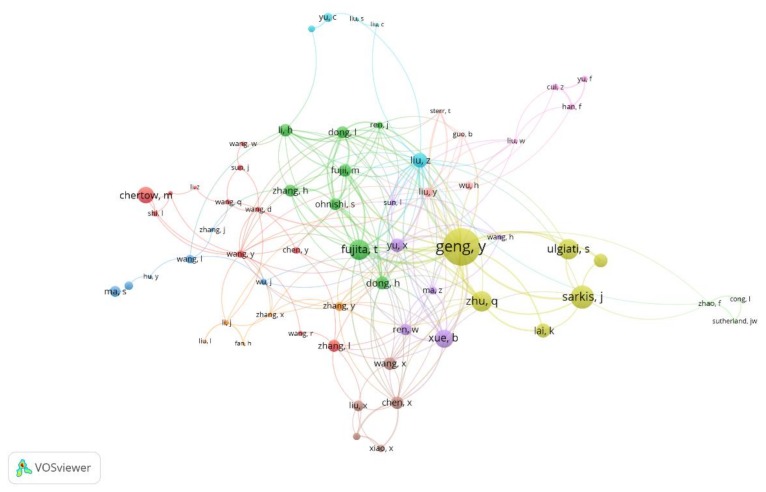
Cocitation map based on bibliographic data.

**Figure 7 ijerph-15-02699-f007:**
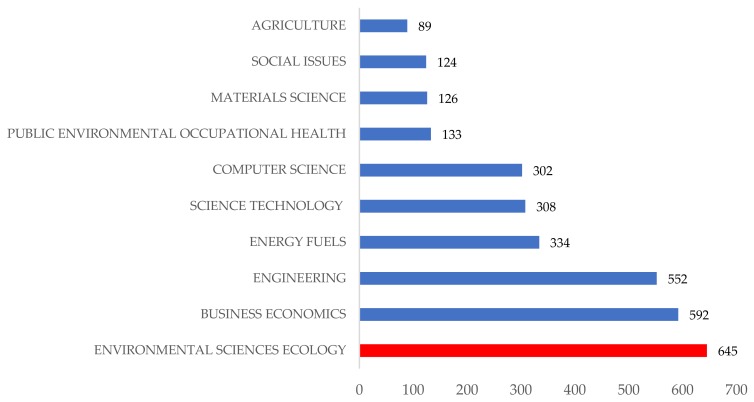
Number of articles published by areas of knowledge.

**Figure 8 ijerph-15-02699-f008:**
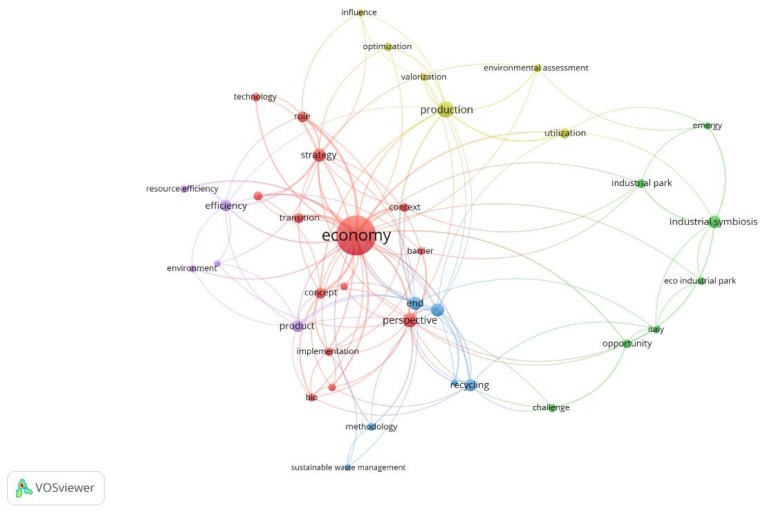
Map based on co-occurrence on the authors keywords from 2014 to 2017.

**Table 1 ijerph-15-02699-t001:** Ranking of countries attending the highest number of articles and citations.

Ranking	Country	Articles	Citations	TC/Art *	h-Index
1	China	201	3728	17.75	30
2	U. Kingdom	102	1687	16.54	19
3	Italy	80	1206	15.08	15
4	Netherlands	64	1453	22.7	19
5	Germany	60	541	9.02	13
6	United States	53	1609	30.36	17
7	Sweden	40	594	14.85	10
8	Spain	39	272	6.97	8
9	Denmark	36	366	10.17	11
10	Australia	33	324	9.82	11
11	Japan	29	779	26.86	15
12	Finland	28	216	7.71	8
13	France	25	165	6.6	8
14	Belgium	20	232	11.6	8
15	Greece	17	203	11.94	7
16	Romania	17	36	2.12	3
17	Brazil	16	38	2.38	4
18	Portugal	16	56	3.5	4
19	Canada	14	364	26	8
20	Austria	12	128	10.67	5
21	Poland	12	128	10.67	5
22	Norway	11	722	65.64	7
23	South Korea	11	314	28.55	7
24	Taiwan	11	146	13.27	5
25	Switzerland	7	137	19.57	5

* Total citations per article.

**Table 2 ijerph-15-02699-t002:** Languages used in scientific articles about circular economy and environment.

Ranking	Languages	N. Articles
1	English	733
2	Spanish	5
3	French	4
4	German	3
5	Others	5

**Table 3 ijerph-15-02699-t003:** Journals and impact.

Ranking	Journal	Articles	Citations	TC/Art *	h-Index
1	Journal of Cleaner Production	140	3439	24.56	28
2	Sustainability	39	253	6.49	9
3	Advanced Materials Research	36	20	0.56	2
4	Resources Conservation and Recycling	36	645	17.92	13
5	Procedia CIRP	33	104	3.15	6
6	Journal of Industrial Ecology	31	607	19.58	12
7	Waste Management	20	164	8.20	5
8	Waste Management Research	19	70	3.68	5
9	Energy Procedia	15	33	2.20	4
10	Applied Mechanics and Materials	14	4	0.29	1
11	24th CIRP Conference on Life Cycle Engineering	13	30	2.31	3
12	Bioresource Technology	12	248	20.67	8
13	Waste and Biomass Valorization	11	23	2.09	3
14	9th CIRP Industrial Product Service Systems IPSS Conference	9	17	1.89	2
15	IOP Conference Series Materials Science and Engineering	8	3	0.38	1
16	Journal of Environmental Management	8	106	13.25	4
17	Renewable Sustainable Energy Reviews	8	277	34.63	7

* Total citations per article.

**Table 4 ijerph-15-02699-t004:** The top ten authors.

Ranking	Author	Articles	Citations	TC/Art *	h-Index
1	Geng, Y.	38	1484	39.05	19
2	Fujita, T.	12	461	38.42	11
3	Zhu, Qh	12	491	10.92	9
4	Liu, Z.	10	219	21.9	8
5	Sarkis, J.	10	587	58.7	10
6	Dong L.	7	173	24.71	6
7	Ulgiati, S.	7	445	63.57	4
8	Xue, B	7	352	50.29	7
10	Zhou, ZF	6	37	6.17	4

* Total citations per article.

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
