# Peer review of "Worldwide Research on Circular Economy and Environment: A Bibliometric Analysis"

_ijerph, 2018, doi:10.3390/ijerph15122699_

Round 1

Reviewer 1 Report

The paper is interesting since is focus on a very relevance topic: the relationship between circular economy and environment. The introduction includes a good   review of the literature with respect to the analyzed topics, contextualizing   the work and facilitating the reading of the paper. The main topics of the   work are clearly defined with the literature review carried out. The   methodology is explained sufficiently in the work, incorporating details that   favor the follow-up of the work. The study is consistent and provides   interesting aspects to consider. The structure of this work is good, and the   paper is well written, what makes it easier to read and to follow. In   addition, the tables and figures presented are appropriated and help to better understand the text, which is an additional value for this type of analysis.

Author Response

Dear reviewer,

Thank you very much for your kind report and comments.

Your contribution will help us to improve our paper.

Warm regards

Reviewer 2 Report

Thanks for giving me the opportunity to contribute on your work. This is an interesting paper that looks at circular economy. However, some issues that require attention are described below.

 I am not convinced about your approach of grounding your research on the systematic literature review on circular economy without using a specific tool (such as methanalysis for instance) or, at least, to carry out a more accurate analysis of the results. My general feeling it that this is not an original contribution in the present form and the analysis is still insufficiently elaborated.

I would spur your literature review to discuss more on the two following issues: 1. what is your main contribution about the previous literature in this field? 2. Why should such results being considered original?

Taking into account the reviews already pubblished in this field that you have cited in your paper (Merli, Guisellini, etc.) I would challenge you to focus more on the different focus of the previous literature regarding circular economy as it might make you argument much stronger.

 I suggest to compare different focuses and theoretical approach or to investigate this line of inquire from a more innovative point of view.

Based on my previous comments I think that your paper cannot be published in its current form. Once again, thank you for the opportunity to review your work. I personally enjoyed reading your work and I wish you the best of luck in your continued research endeavors.

Author Response

Response to Reviewer 2 Comments

Point 1: I am not convinced about your approach of grounding your research on the systematic literature review on circular economy without using a specific tool (such as methanalysis for instance) or, at least, to carry out a more accurate analysis of the results.

Response 1: First of all, thank you very much for your review and your comments that will help us, for sure, to improve this paper. For this research, we have followed the process for bibliometric research analysis suggested by professors Brereton, Kitchenham, Budgen, Turner, Khalil and Lessons (published in 2007), and the Citation Index for Science (Garfield, 1955). As in many others articles analysing large data sets, we decided to use a bibliometric method, since it facilitates the examination of WOS databases.

We have included some more references in the text, supporting both the circular economy topic and the bibliometric analysis.

Anyway, we have considered your suggestion of meta-analysis, and included it in the conclusion section, as a proposal for future researches, by analysing the topic under a different point of view.

Point 2: My general feeling it that this is not an original contribution in the present form and the analysis is still insufficiently elaborated. I would spur your literature review to discuss more on the two following issues: 1. what is your main contribution about the previous literature in this field? 2. Why should such results being considered original? I would challenge you to focus more on the different focus of the previous literature regarding circular economy as it might make you argument much stronger. I suggest to compare different focuses and theoretical approach or to investigate this line of inquire from a more innovative point of view.

Response 2: We appreciate your comments. Following your suggestions, we have improved the paper by incorporating the explanation of the cluster analysis, in authors section. We have also include a fractional counting method to analyse the most relevant keywords on CE of the last five years, in order to show the trend of research and suggest and highlight the more important relationships on topics. Thus, the article represents a contribution by identifying the main tendencies in circular economy research and environment and, from there, it proposes future research lines.

Although other works have done also a bibliometric analysis of the circular economy, most of them have focus on specific industrial sectors and/or regions (e.g., Türkeli et al. focused on European Union and China; Tisserant et al., on solid waste; Jackson et al., on managing metals). Our paper analysed the worldwide research on circular economy and environment.

We have identified different, such as eco-innovation, eco-design and waste management. Other interesting interactions are those related to energy and industry, remanufacturing and resource efficiency, eco-efficiency and cleaner production. Topics such as LCA, green economy, sustainable development, climate change and environment are also emergent research lines.

Point 3: I personally enjoyed reading your work and I wish you the best of luck in your continued research endeavours.

Response 3: Once again, thank you very much. We are happy you enjoyed this research, and we appreciate very much your notes and comments, so it helps us to improve the paper.

Reviewer 3 Report

Dear authors

Sincerely, your article is very useful for researchers in the field of circular economy and the environment. In this sense, and so that in the future it can be very referenced, I propose some papers to incorporate them to strengthen both the introduction, the methodology and the conclusions.

In relation to the proposals of the papers  that I propose to improve the content of the research, I think that all  of them  can give you ideas to improve your research work,with the aim that in the future your article is a good reference. However, you can use the ones that you consider most important for your research.

Proposals

Keywords: sustainability, cross world research, Global research environment

 Introduction

Circular economy for the built environment: A research frameworkJournal of Cleaner Production, Volume 143, 1 February 2017, Pages 710-718 Francesco Pomponi, Alice Moncaster

Circular economy – From review of theories and practices to development of implementation tools Resources, Conservation and Recycling, Volume 135, August 2018, Pages 190-201Yuliya Kalmykova, Madumita Sadagopan, Leonardo Rosado

Núñez-Cacho, P.; Molina-Moreno, V.; Corpas-Iglesias, F.A.; Cortés-García, F.J. Family Businesses Transitioning to a Circular Economy Model: The Case of “Mercadona”. Sustainability 2018, 10, 538

Methodolgy

Lozano, F. J., Freire, P., Guillén-Gozalbez, G., Jiménez-Gonzalez, C., Sakao, T., Dowell, N. Mac Viveros, T. (2016). New perspectives for sustainable resource and energy use, management and transformation: Approaches from green and sustainable chemistry and engineering. Journal of Cleaner Production, 118, 1–3. http://doi.org/10.1016/j.jclepro.2016.01.041

The Global Research Environment.Chapter 3 Author links open overlay panelPamela F.Miller https://doi.org/10.1016/B978-0-12-805059-0.00003-1

Molina-Sánchez, E.; Leyva-Díaz, J.C.; Cortés-García, F.J.; Molina-Moreno, V. Proposal of Sustainability Indicators for the Waste Management from the Paper Industry within the Circular Economy Model. Water 2018, 10, 1014.

Niesten, E., Jolink, A., Beatriz, A., Sousa, L. De, Chappin, M., & Lozano, R. (2017). Sustainable collaboration : The impact of governance and institutions on sustainable performance. Journal of Cleaner Production, 155, 1–6. http://doi.org/10.1016/j.jclepro.2016.12.085

Nuñez-Cacho, P.; Górecki, J.; Molina-Moreno, V.; Corpas-Iglesias, F.A. What Gets Measured, Gets Done: Development of a Circular Economy Measurement Scale for Building Industry. Sustainability 2018, 10, 2340. 

 Conclusions

Lahti, T.; Wincent, J.; Parida, V. A Definition and Theoretical Review of the Circular Economy, Value Creation, and Sustainable Business Models: Where Are We Now and Where Should Research Move in the Future? Sustainability 2018, 10, 2799.

Domingues, A. R., Lozano, R., Ceulemans, K., & Ramos, T. B. (2017). Sustainability reporting in public sector organisations: Exploring the relation between the reporting process and organisational change management for sustainability. Journal of Environmental Management, 192, 292–301. http://doi.org/10.1016/j.jenvman.2017.01.074

Molina-Moreno, V.; Leyva-Díaz, J.C.; Llorens-Montes, F.J.; Cortés-García, F.J. Design of indicators of circular economy as instruments for the evaluation of sustainability and efficiency in wastewater from pig farming industry. Water 2017, 9, 653.

Author Response

Response to Reviewer 3 Comments

Point 1: Sincerely, your article is very useful for researchers in the field of circular economy and the environment. In this sense, and so that in the future it can be very referenced.

Response 1: First of all, thank you very much for your review and your comments that will help us, for sure, to improve this paper

Point 2: Keywords: sustainability, cross world research, Global research environment

Response 2: Thanks so much. The keywords have been included.

Point 3: I propose some papers to incorporate them to strengthen both the introduction, the methodology and the conclusions. In relation to the proposals of the papers  that I propose to improve the content of the research, I think that all  of them  can give you ideas to improve your research work, with the aim that in the future your article is a good reference. However, you can use the ones that you consider most important for your research.

Response 3: We appreciate your comments. Following your suggestions we have included  some of the references you proposed. After a review of the references recommended, we have consider to include specifically this papers that support our contribution:

·        Circular economy for the built environment: A research framework Journal of Cleaner Production, Volume 143, 1 February 2017, Pages 710-718 Francesco Pomponi, Alice Moncaster

·        Núñez-Cacho, P.; Molina-Moreno, V.; Corpas-Iglesias, F.A.; Cortés-García, F.J. Family Businesses Transitioning to a Circular Economy Model: The Case of “Mercadona”. Sustainability 2018, 10, 538

·        Molina-Sánchez, E.; Leyva-Díaz, J.C.; Cortés-García, F.J.; Molina-Moreno, V. Proposal of Sustainability Indicators for the Waste Management from the Paper Industry within the Circular Economy Model. Water 2018, 10, 1014.

·        Nuñez-Cacho, P.; Górecki, J.; Molina-Moreno, V.; Corpas-Iglesias, F.A. What Gets Measured, Gets Done: Development of a Circular Economy Measurement Scale for Building Industry. Sustainability 2018, 10, 2340.

·        Domingues, A. R., Lozano, R., Ceulemans, K., & Ramos, T. B. (2017). Sustainability reporting in public sector organisations: Exploring the relation between the reporting process and organisational change management for sustainability. Journal of Environmental Management, 192, 292–301.

·        Molina-Moreno, V.; Leyva-Díaz, J.C.; Llorens-Montes, F.J.; Cortés-García, F.J. Design of indicators of circular economy as instruments for the evaluation of sustainability and efficiency in wastewater from pig farming industry. Water 2017, 9, 653.

Round 2

Reviewer 2 Report

I can confirm that changes have been conducted. The authors made effort to clearly answer the research questions and all requirements from all the reviewers.

In my opinion the paper could be published in the present form.

Reviewer 3 Report

Dear authors,

I have read your work again and, from my point of view, I have incorporated the improvements so that it can be published.

Please, I encourage you to continue working in this interesting field of research.